# Rate-Informed Discovery via Bayesian Adaptive Multifidelity Sampling

**Aman Sinha**[*]**, Payam Nikdel**[*]**, Supratik Paul, and Shimon Whiteson**
Waymo, LLC
{thisisaman, payamn, supratikpaul, shimonw}@waymo.com

**Abstract:** Ensuring the safety of autonomous vehicles (AVs) requires both accurate estimation of their performance and efficient discovery of potential failure cases. This paper introduces Bayesian adaptive multifidelity sampling (BAMS), which leverages the power of adaptive Bayesian sampling to achieve efficient discovery while simultaneously estimating the rate of adverse events. BAMS prioritizes exploration of regions with potentially low performance, leading to the identification of novel and critical scenarios that traditional methods might miss. Using real-world AV data we demonstrate that BAMS discovers 10 times as many issues as Monte Carlo (MC) and importance sampling (IS) baselines, while at the same time generating rate estimates with variances 15 and 6 times narrower than MC and IS baselines respectively.

**Keywords:** Autonomous Driving, Rare-event Simulation, Adaptive Sampling

## 1 Introduction

Commercial autonomous vehicle (AV) development usually follows an iterative process wherein issues with the current planner are identified, root causes are determined, and the issues are addressed for the next release. While these software updates are meant to improve performance, they can also lead to regressions. Evaluating the long tail of safety issues is particularly difficult: as the AV's performance improves, safety-related issues become rarer and more difficult to discover. In turn, this makes improving the planner increasingly challenging, as identifying failure cases is the first step towards addressing them. Together with discovering particular failure modes, it is also important to efficiently estimate the overall safety or failure rate of any potential release before deployment.

Improvements to an AV planner can be tested on the road with human drivers in the vehicle ready to take over in the event of imminent danger. However, this approach is expensive and does not ensure sufficient coverage of driving scenarios since we cannot control the environment [1]. Furthermore, since this approach essentially produces a Monte Carlo estimate of the AV's performance, the scale required for evaluation near human-level performance is too large [2] to repeat at regular intervals during development. As such, simulation plays a key role in industrial AV software evaluation (see, e.g., [3, 4]), as it enables testing software in diverse settings, at a fraction of the cost and time, and with no risk to the public. However, a naive simulation-based approach to evaluation also suffers from intractable scaling and can be prohibitively expensive.

In this paper, we address the twin problems of (i) estimating the safety of a given AV planner and (ii) discovering failure cases that can then be used to drive further improvements. Concretely, given a distribution $P$ of simulation parameters that describe the AV's operational design domain, the governing problem for evaluating safety is to estimate the rate of adverse events:

$$p_\gamma := \mathbb{P}(f(X) \leq \gamma) = E_X \left[ \mathbf{1}\{f(X) \leq \gamma\} \right], \tag{1}$$

where $X \sim P$, the function $f : \mathcal{X} \to \mathbb{R}$ is a performance metric scoring the realization $x \in \mathcal{X}$ of the AV and its environment, and $\gamma$ is a user-defined threshold below which behavior is undesirable. Estimating $p_\gamma$ is a rare-event simulation problem, so naive Monte Carlo methods are expensive and we must resort to more tractable IS techniques; these techniques place high probability on regions of $\mathcal{X}$ where $f(x)$ is small.

However, simply estimating $p_\gamma$ (the objective of importance sampling (IS)) is insufficient, as simulation testing must provide actionable feedback on how to improve AV design (Figure 1). This feedback comes in the form of concrete examples $x_i$ where $f(x_i) \leq \gamma$. In particular, during the continuous

---

[*]Equal contribution

8th Conference on Robot Learning (CoRL 2024), Munich, Germany.

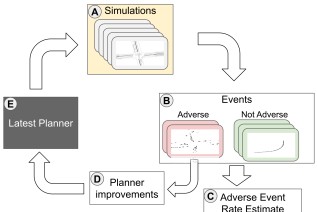

**Figure 1:** Rate-informed discovery loop – **A**: A diverse set of simulations are run with the latest version of the AV planner. **B**: Some simulations lead to adverse events which are attributable to the planner. **C**: The results from all simulations are used to compute an unbiased estimate of the rate of adverse events. **D**: The adverse events found in **B** are used to understand and address the failure cases. **E**: An improved planner is deployed, and the cycle repeats.

feedback loop of simulation testing followed by design improvement, it is most useful for simulation to discover failure cases $x_i$ with a) relatively high likelihood under $P$ and b) novelty with respect to previously discovered failure examples. From a practical perspective, prioritization by likelihood is obvious when trying to allocate resources to fixing problems, and novelty is important because previously-discovered failure modes typically have existing work underway to improve behavior.

We define *rate-informed discovery* as the problem of returning novel, likely examples of undesirable AV behavior in service of confidently estimating the rate $p_\gamma$. Our goal is to perform rate-informed discovery as efficiently as possible. Just as with purely estimating the rate $p_\gamma$, naive Monte Carlo is intractably expensive for rate-informed discovery. Furthermore, IS techniques that rely heavily on previously determined failure modes are poor at discovering novel issues. In this paper, we develop *Bayesian adaptive multifidelity sampling* (BAMS), which adaptively samples from $\mathcal{X}$ in a targeted fashion and builds an importance sampler for estimating $p_\gamma$. As efficiency is paramount, we exploit the availability of a simulation hierarchy: low-fidelity yet cheap simulators as well as high-fidelity but expensive simulators that model real-world environments as accurately as possible. As we show below, this *multifidelity* simulation setup enables BAMS to perform better rate-informed discovery for any given simulation budget than using the high-fidelity simulator alone.

## 2 Background and related work

**Gaussian processes** A Gaussian process (GP) [5] imposes a prior belief on an unknown random function $\theta : \mathcal{X} \subseteq \mathbb{R}^d \to \mathbb{R}$ through its mean function $\mu(x)$ (for convenience usually set to zero) and covariance function $k(x, x')$, such that any finite collection of function values $\{\theta(x_i)\}$ has a joint Gaussian distribution. A popular choice of the covariance function is the Matérn kernel: $k(x, x') \propto t^\nu K_\nu(\sqrt{2\nu}t)$, where $\nu$ is a lengthscale parameter (typically $\nu = 2.5$), $K_\nu$ is the modified Bessel function of the second kind, and $t := \left\{ (x - x')^T \Sigma^{-1} (x - x') \right\}$ is a distance scaled by a diagonal matrix $\Sigma$ of lengthscales for each dimension. Letting $X_n \in \mathbb{R}^{n \times d}$ be the matrix of observed inputs and $\theta(X_n)$ be the vector of associated function values, the posterior prediction $\theta(x^*)$ for an input $x^*$ is a Gaussian distribution $\mathcal{N}(\mu_n(x^*), \sigma_n^2(x^*))$:

$$\mu_n(x^*) = k(x^*, X_n) k(X_n, X_n)^{-1} \theta(X_n),$$
$$\sigma_n^2(x^*) = k(x^*, x^*) - k(x^*, X_n) k(X_n, X_n)^{-1} k(X_n, x^*).$$

The posterior covariance between any two inputs $(x, x')$ is $\text{cov}_n(x, x') = k(x, x') - k(x, X_n) k(X_n, X_n)^{-1} k(X_n, x')$. There is a vast literature on speeding up GP inference, namely assuaging the $O(n^3)$ time complexity of inverting $k(X_n, X_n)$. In particular, inducing point approximations [6, 7, 8] impose structure on the GP prior, which in turn induces (block) sparsity onto $k(X_n, X_n)$. Whereas inducing point methods make posterior mean inference more tractable, other approaches make predictive variances cheaper by exploiting sparsity in the underlying data and employing iterative techniques for approximations [9, 10]. All of these approaches are motivated by the regime wherein the training set size $n$ is typically much larger than the test set. Our setting is exactly the opposite; the discovery process occurs frequently in AV development cycles, so it is highly budget-constrained and $n$ is small. In our experiments, we simulate and use only tens of samples for the GP posterior, whereas our predictive covariance is computed on a much larger set of available points. Thus, while our approach is complementary to the existing literature and can be used in conjunction with these methods, we do not benefit substantially from them.

**Bayesian optimization and quadrature** Bayesian optimization (BO) [11, 12] and quadrature (BQ) [13, 14, 15] use GPs for targeted function evaluation to perform minimization for expensive functions or to compute expensive integrals over $\mathcal{X}$. BO and BQ methods utilize an *acquisition function* to select points to evaluate. Acquisition functions typically balance exploration of uncertain regions in $\mathcal{X}$ with exploitation of previous evaluations $f(x)$ (see, e.g., Wilson et al. [16] for an overview).

**Multifidelity simulation** There is a rich literature on using multiple fidelities for evaluation, also known as surrogate modeling (see, e.g., [17, 18, 19, 20, 21]). Specializing to GPs, Bonilla et al. [22]

introduce the concept of multi-task GP prediction, which has subsequently been employed in BO use cases [23, 24, 25]. The multi-task covariance kernel is highly structured as a tensor product of a base kernel and a covariance between discrete tasks [22]. By contrast, Marco et al. [26] and Poloczek et al. [27] consider modeling the dependence between fidelities via additive noise GPs. We employ this modeling technique in BAMS due to its generality; it requires little prior knowledge about the relationship between multiple simulators and allows us to easily incorporate costs of each fidelity into our acquisition function, especially when multiple fidelities are available for the same data point $x$.

**AV safety-measurement techniques** AV evaluation literature generally considers approaches for formal verification [28, 29, 30, 31], (probabilistic) falsification [32, 33, 34], and rate estimation [35]. Formal verification aims to define and then prove correctness of behavior; this is both NP hard [36] and rife with subjectivity on notions of correctness. Falsification aims to find any problems with the AV system under test, and probabilistic variants aim to find high-likelihood failures [33]. Finally, rate estimation aims to measure the probability of adverse events under a (possibly empirically defined) distribution of driving environments [35]. Previous papers note a dichotomy between falsification and rate estimation [37, 38], which our paper aims to bridge; our rate-informed discovery loop first evaluates targeted samples solving an optimization problem (similar to falsification), and we follow suit with updating an importance sampler to measure rates. Unlike many probabilistic falsification approaches, which require whitebox access to the system [34], our formalism performs optimization with entirely blackbox access. See Appendix F for a detailed discussion on how our Bayesian approach connects to existing rare-event simulation techniques, addresses key challenges within this domain, and contributes to the broader field of AV safety evaluation.

## 3 Bayesian adaptive multifidelity sampling

In typical AV development environments, the distribution $P$ over which we compute the rate of adverse events (1) is an empirical distribution over *run segments*—specified time intervals (e.g., 30 seconds) of logged real-world driving over which simulations are performed [39, 40]. We consider the empirical distribution over $N$ run segments, and each run segment is encoded by an embedding $x \in \mathbb{R}^d$. There are a variety of ways to create such an embedding (see, e.g., [41, 42, 43, 39, 44, 45]); BAMS does not have any restrictions for this embedding, although smaller dimensionality $d$ improves computational efficiency. We provide details for our embedding model in Section 4.

Our formalism begins by first considering a Bayesian estimator for the rate of adverse events (1). Specifically, we define a random function $\theta : \mathcal{X} \to \mathbb{R}$ and impose a GP prior over it such that the collection $\{\theta(x_i)\}$ for any subset of run segments $\{(x_i)\}$ has a joint Gaussian distribution. Considering a set of $n$ evaluated run segments and their associated performance metrics $\mathcal{X}_n := \{(x_i, f(x_i))\}_{i=1}^n$, we define the shorthand $\theta_n$ and $E_{\theta_n}[\cdot]$ as the posterior GP and the expectation with respect to this posterior induced by conditioning on the $\sigma$-algebra generated by $\mathcal{X}_n$. We can then define a function which models the presence of adverse events based on our GP posterior:

$$g(X, \theta_n) := \mathbf{1}\{\theta_n(X) \leq \gamma\} \in \{0, 1\}, \tag{2}$$

with marginals $\hat{p}_\gamma(\theta_n) := E_X[g(X, \theta_n)]$ and $p_n(x) := E_{\theta_n}[g(x, \theta_n)]$. The marginal $\hat{p}_\gamma(\theta_n)$ is an estimator of the rate of adverse events (1).

Using $\theta$ to model the performance metric $f(x)$ of a run segment rather than the binary indicator $\mathbf{1}\{f(X) \leq \gamma\}$ has two benefits: (i) modeling the real-valued function provides better signal for the GP to interpolate, and (ii) in typical AV development paradigms, we want to understand the safety performance with respect to multiple values of the threshold $\gamma$, so we can reuse a single trained GP model. Next we describe various aspects of BAMS, motivated by computational efficiency.

**Acquisition function** As noted in Section 2, the acquisition function defines a minimization objective that selects the next point(s) to sample the performance metric $f(x)$ and compute the new GP posterior. Our goal of rate-informed discovery implies that choosing evaluation points to minimize the variance of the estimator, $\mathrm{Var}(\hat{p}_\gamma(\theta))$, is beneficial. In particular, minimizing $\mathrm{Var}(\hat{p}_\gamma(\theta))$ provides a confident estimate of the probability (1). Furthermore, the calculation of $\mathrm{Var}(\hat{p}_\gamma(\theta))$ is dominated by regions in $\mathcal{X}$ that have both high likelihood under $P$ and high uncertainty under $\theta_n$. Thus, minimizing $\mathrm{Var}(\hat{p}_\gamma(\theta))$ prioritizes sampling points $x$ that are clustered near other points (high likelihood) and are far away from previously sampled points (high uncertainty)—precisely our goal for rate-informed discovery. However, this variance requires an expectation over the joint distribution $X, X' \overset{\text{i.i.d}}{\sim} P$ (see Appendix B.1); because $P$ is the empirical distribution over run segments, this $O(N^2)$ computation is too expensive. Instead, we consider a cheaper upper bound.

First we define the point-variance function $h_n(x) := p_n(x)(1 - p_n(x))$. The average of $h_n(x)$ over $P$, which requires only $O(N)$ computation, is greater than the variance of the GP estimator:

**Proposition 3.1.** *The variance of the estimator $\hat{p}_\gamma(\theta_n)$ is upper-bounded by the average point variance:* $\mathrm{Var}(\hat{p}_\gamma(\theta_n)) \le E_X[h_n(x)]$.

See Appendix B.2 for the proof. Since we evaluate the next $m$ points in a batch setting, the order of these points does not matter, and our acquisition function is a function of the set $\{x_j\}_{n+1}^{n+m}$. In particular, consider the expectation of the point-variance function conditioning on the set of locations of these $m$ future evaluations:

$$\beta_n\left(x; \{x_j\}_{n+1}^{n+m}\right) := E_{\theta_n}\left[h_{n+m}(x) | \{X_j = x_j\}_{n+1}^{n+m}\right]. \tag{3}$$

This forward-looking point variance has a nested expectation $E_{\theta_n}\left[E_{\theta_{n+m}}\left[ \cdot \ | \{X_j = x_j\}_{n+1}^{n+m}\right]\right]$, the average over all future posteriors conditioned on the locations $\{x_j\}_{n+1}^{n+m}$; this is an analytic computation in GP models. Then, our acquisition function to choose the next $m$ evaluation locations is the average over $P$ of this forward-looking point variance:

$$J\left(\{x_j\}_{n+1}^{n+m}\right) := E_X\left[\beta_n\left(X; \{x_j\}_{n+1}^{n+m}\right)\right]. \tag{4}$$

An immediate corollary of Proposition 3.1 is that this acquisition function is an upper bound on the expected variance conditioned on the same points (see Appendix B.3 for the proof):

**Corollary 3.2.** *The expected variance of $\hat{p}_\gamma(\theta_n)$ conditioned on locations $\{x_j\}_{n+1}^{n+m}$ of the next evaluation points is upper-bounded by the acquisition function:*

$$E_{\theta_n}\left[\mathrm{Var}(\hat{p}_\gamma(\theta_{n+m})) | \{X_j = x_j\}_{n+1}^{n+m}\right] \le J\left(\{x_j\}_{n+1}^{n+m}\right).$$

**Sequential selection** For $m, n \ll N$, the acquisition function (4) can be efficiently calculated in $O(m^2 N)$ time [46] (see Appendix A.2 for details). Because our distribution $P$ is the empirical distribution over $N$ samples, jointly choosing the locations for $m > 1$ points is a combinatorial problem over $\binom{N}{m}$ combinations. Since $\binom{N}{m} = O(N^m/m^m)$, this combinatorial problem has an overall time complexity of $O(m^{2-m} N^{m+1})$, which is too expensive. As a consequence, rather than choose all $m$ locations jointly, we instead choose each one sequentially. Naively, this sequential selection takes $O(m^3 N^2)$ time, but exploiting the recursive structure of the problem allows us to reduce this cost to $O(m N^2)$ (see Appendix A.3 for details). In contrast, sequential selection using expected variance as an acquisition function requires $O(m N^3)$ time.

In the sequential setup, our minimization objective for each iteration $n + 1 \le i \le n + m$ is:

$$\underset{x}{\text{minimize}} \ \ J\left(\{x_j\}_{n+1}^{i-1} \cup \{x\}\right) - J\left(\{x_j\}_{n+1}^{i-1}\right),$$

where $\{x_j\}_{n+1}^n := \emptyset$ and $J(\emptyset) := E_X[h_n(x)]$. The subtraction of the objective from the previous iteration is not necessary for this minimization problem but it is useful to define our problem this way to motivate the corresponding multifidelity problem below.

**Multifidelity sampling** Thus far, we have considered a model wherein there is only one way to perform evaluation on a selected point $x$ via a single simulator. However, in our AV use case, we often have multiple simulators whose computational cost correlates with fidelity. For example, instead of simulating the entire AV planner and other traffic participants, we can simulate purely with non-interactive traffic participants or a distilled version of the AV planner. Intuitively, a noisier but cheaper simulator that correlates well with the high-fidelity simulator offers an information advantage under budget constraints: we can learn more about the ground truth with the same simulation cost. Of course, we also need to learn the correlation between simulators without incurring extra overhead. To do so, we augment our Bayesian generative model to include multiple fidelities and modify the acquisition function (4) to account for the trade-off between fidelity and computational cost.

For $l \in \{0, 1, \ldots, L\}$, let the fidelities of simulation be denoted by $f_l$, with the original high-fidelity ($l = 0$) simulator denoted $f_0$. Consider the augmented input $y_l := (x, l)$, which then allows us to define the augmented simulation function $f(y_l) := f_l(x)$. Next, let the model for each augmented input $\theta : \mathcal{X} \times \mathbb{N} \to \mathbb{R}$ be defined as $\theta(y_l) := \theta_0(x) + \alpha_l(x)$, where $\theta_0$ is the GP model over the function $f_0$, $\alpha_0 := 0$, and for $l \ge 1$, $\alpha_l$ is an independent zero-mean GP with covariance function $k_l$. Then, the augmented function $\theta$ is a GP with mean $\theta(y_l) = \mu(x)$; the covariance between points $y_a = (x, a)$ and $y_b' = (x', b)$ is: $k(y_a, y_b') = \mathrm{Cov}(\theta_0(x) + \alpha_a(x), \theta_0(x') + \alpha_b(x')) = k(x, x') + \delta_{ab} k_a(x, x')$, where $\delta(a, b)$ is the Kronecker delta function.

To update the acquisition function (4), we first define $c(y_l)$ as the cost associated with evaluating each simulator $f_l$. In typical AV settings, this cost is expressed in units of money or flops of computation.

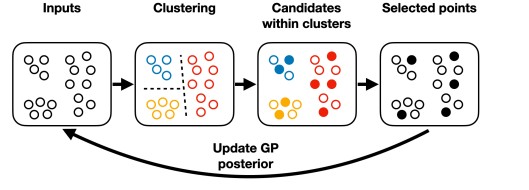
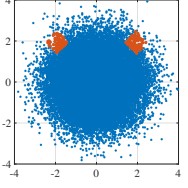
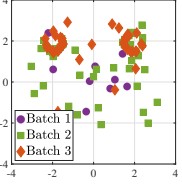

| **(a)** Schematic of BAMS algorithm | **(b)** Synthetic data | **(c)** Selected points |

**Figure 2:** Illustration of BAMS. a) Iterative loop of our approach, as defined in Algorithm 1 (Appendix A.1). In each iteration, we separate the inputs into clusters, solve problem (5) over each cluster, and then select the final points from the candidates in all the clusters. After performing simulations over these points, we update the GP posterior. For simplicity, we neglect illustrating multiple fidelities in this graphic. b) Distribution of samples $P$ for a simple synthetic setting. Samples are colored red if $f(x) \leq \gamma$ and blue otherwise. c) Points selected over three iterations (batches) of our algorithm in the synthetic setting. Initial exploration over the state space in the first iteration transitions to targeted uncertainty reduction around the regions of interest in the third iteration.

We define $c(y_0) := 1$ and by construction $c(y_l) < 1$ for $l \geq 1$. Whereas in the single-fidelity setting we select a batch of $m$ points, we now consider selecting $r \geq m$ points whose total cost is $m$. Then, the new multifidelity acquisition problem is to minimize the cost-normalized expected forward-looking point-variance function. For clarity, we write $y$ in its expanded form $(x, l)$:

$$\underset{(x,l)}{\text{minimize}} \quad \frac{1}{c(x,l)} J\left(\{(x,l)_j\}_{n+1}^{i-1} \cup \{(x,l)\}\right) - J\left(\{(x,l)_j\}_{n+1}^{i-1}\right), \tag{5}$$

where $n+1 \leq i \leq n+r$. Unlike in the single-fidelity case, the subtraction of the previous iteration's objective is now required—we normalize the *decrease* of the forward-looking point-variance with respect to the price of simulation.

The theoretical results above extend directly to the mutifidelity setting where we consider $P$ as the empirical distribution over all inputs and all fidelities. In general, we may not have the ability to evaluate every input $x_i$ at every level $l$, in which case we simply define our distribution $P$ over the empirical distribution of locations and corresponding available simulation fidelities. In Section 4, we assume that all simulators are available for all $N$ points.

**Clustering** As noted in Section 2, our problem setting is different from most work on GPs and BO/BQ that are concerned with computational efficiency. Specifically, the number of training data points is extremely small (typically tens in our experiments) compared to $N$. Instead, our bottleneck is the sequential selection of points to evaluate via minimization of the acquisition function, an $O(mN^2)$ computation. To decrease this cost, we employ clustering. With $S$ clusters, the complexity of our sequential selection strategy is $O(mN^2/S^2)$. We can trivially take advantage of parallel computation in this approach as well. See Appendix A.4 for details.

The complete BAMS algorithm with clustering is detailed in Algorithm 1 (Appendix A.1). Figure 2 illustrates our approach on a synthetic two-dimensional dataset. We show a schematic of Algorithm 1 in Figure 2a. Figure 2b visualizes the input distribution $P$ and Figure 2c shows samples selected by our algorithm. For a detailed analysis of the synthetic experiment, see Appendix D.

## 4 Experiments

To showcase the effectiveness of BAMS, we apply it to a realistic AV context using the Waymo Open Sim Agent Challenge (WOSAC) dataset [40]. The WOSAC dataset contains $N = 44,911$ logged run segments and 32 rollouts per run segment generated via AV planner simulation. The AV planner resimulates each segment by predicting the future states of all agents, conditioned on the initial states of the logged run segment. Our goal is to estimate the rate of near-accident events. We describe all aspects of our setup, including the embedding model to generate $x$ for each run segment, the performance metric $f(x)$, experimental procedure, multifidelity setup, and evaluation criteria.

**Embedding model** The input embedding is generated in two stages:

1. 512-dimensional embedding: Using a dataset of run segments, we extract features such as road layouts, other road users' positions, and AV kinematics. These features are used to generate top-down images, encoded into 512-dimensional vectors using a convolutional neural network. Training uses contrastive learning on image pairs labeled as being from the same run segment or not. See Bronstein et al. [39] for details.

2. Dimensionality reduction: We train an encoder-decoder model to reduce the embedding to a 12-dimensional representation, which is more computationally tractable for GPs.

Since the 32 AV planner rollouts in WOSAC are generated based on one second of history, our embedding model utilizes the embedding at the start of the run segment.

**Performance metric** We utilize the 32 rollouts to calculate a single time-to-collision (TTC) value for each run segment using the formula $f(x) = Q_{0.25}(mTTC_{[1]}(x) \dots mTTC_{[32]}(x))$, where $mTTC_{[i]}(x)$ is the minimum TTC across the $i^{\text{th}}$ rollout for run segment with embedding $x$ and $Q_{0.25}(\dots)$ is the $25^{\text{th}}$ percentile of the arguments. We use the $25^{\text{th}}$ percentile to be robust to outliers in the rollouts. Finally, we set the rate $p_\gamma$ at 1% by using the threshold $\gamma = 0.43$ for this metric.

**Experimental procedure** We utilize BAMS with $S = 6$ clusters, based on the parameter-tuning study presented in Appendix C.1. We compare BAMS, which leverages both fidelity levels, with its ablated counterpart *Bayesian adaptive sampling* (BAS), which only employs the high-fidelity simulator. Additionally, we measure performance of BAMS and BAS against the following baselines:

- MC: Standard Monte Carlo sampling.
- MC-GP: Our method with random sampling as the acquisition function.
- MCM-GP: Multifidelity version of MC-GP.
- DS: IS with *difficulty-model* scores.
- DS-GP: Our method with IS based on *difficulty-model* scores as the acquisition function.
- CE: Cross-entropy method [47]. See Appendix E for implementation details.

DS and DS-GP leverage IS based on a *difficulty model*. Inspired by Bronstein et al. [39], this model predicts the intrinsic difficulty of a run segment based on previous simulations conducted under diverse planner versions. Our experimental procedure for all models is as follows:

*A. Initialization*: All methods begin with a first batch of random samples with a budget of $m_1 = 20$ evaluations. This ensures unbiased exploration of the search space. Subsequently, the GP-based methods leverage this initial data to train their hyperparameters by maximizing the marginal log-likelihood (MLL) using the Adam optimizer [48].

*B. Two batches with budget $m_b$*: Each method samples two additional batches, each with a budget of $m_b = 15$ evaluations. We retrain the GP hyperparameters after each batch.

**Multifidelity setup** For BAMS and MCM-GP, we utilize two fidelity levels:

- Level 0 (high fidelity): This level uses the $25^{\text{th}}$ percentile TTC calculated from all 32 rollouts $Q_{0.25}(mTTC_{[1]} \dots mTTC_{[32]})$ with cost $c(y_0) = 1.0$.
- Level 1 (low fidelity): This level uses the $25^{\text{th}}$ percentile TTC calculated from only 5 randomly selected rollouts, denoted $Q_{0.25}(mTTC_{[1]} \dots mTTC_{[5]})$, whereby $c(y_1) = 5/32$. This parameter was chosen based on a parameter-tuning study (see Appendix C.3) conducted over $Q_{0.25}(mTTC_{[1]} \dots mTTC_{[i]})$ with corresponding cost $i/32$, evaluating the model's recall at a retention budget of $5p_\gamma N$.

## 4.1 Evaluation criteria

We evaluate all approaches using two criteria: retention-recall curves and rate estimation/discovery via IS, detailed below.

**Retention-recall curves** At the end of each batch, we use retention-recall curves to compare how effectively different methods identify novel failure modes. These curves offer insights into how efficiently each method recalls failure modes while minimizing the number of retained samples. For GP methods, the curve is generated by sorting the high-fidelity data points ($l = 0$) based on $p_n(x_i)$ estimates. For MC, samples are sorted randomly. Recall is then calculated at different retention levels, representing the fraction of failures found within the top-ranked samples.

To facilitate interpretability, we scale the $x$-axis (retention) of retention-recall curves by $p_\gamma N$ (number of failures). Thus, a value of 1 on the $x$-axis corresponds to retaining the $p_\gamma N$ highest-scoring samples. This scaling also allows us to infer precision. For example, if recall is $100\%$ at a retention of $2p_\gamma N$, all failure modes are captured within the top $2p_\gamma N$ sample scores $p_n(x_i)$, or precision is 0.5.

Retention-recall curves illustrate the trade-off between recall and the number of retained samples, which reveals a method's ability efficiently discovering failure modes. However, generation of this curve requires evaluating all $x_i$ in the high-fidelity simulator, so it is not a realistic way of measuring rate-informed discovery in AV development settings. We consider tractable statistics next.

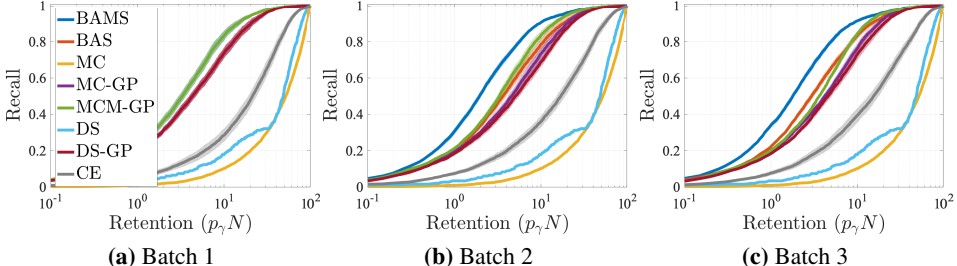

**(a)** Batch 1      **(b)** Batch 2      **(c)** Batch 3

**Figure 3:** Retention-recall curves for AV experiments. The $x$-axis represents the retention budget scaled by $p_\gamma N$ while the $y$-axis shows the recall at each corresponding retention budget.

**Rate estimation and discovery via IS** After completing the third batch of simulation, we conduct a combined evaluation of both rate estimation and discovery. We do 200 repeated trials using model scores to perform IS with a budget of $K = 5p_\gamma N$ high-fidelity samples ($l = 0$). We calculate the resulting recall and relative variance (RV) of the rate estimate, defined as the variance divided by $p_\gamma^2$ (e.g., the square of the coefficient of variation). For GP methods, we define our importance sampler as $p_n(x)^\alpha$, with $\alpha = 2.5$ chosen to minimize relative variance (see Appendix C.2). For MC, sampling is based on random scores.

This IS approach guarantees that our estimate of the failure rate $p_\gamma$ is unbiased by construction, even if our underlying GP model $\theta$ is biased. Together, these two statistics—relative variance and recall—test each method's ability to perform rate-informed discovery: a good sampler has high recall and low variance for the rate estimate. Most importantly, generation of these statistics is a natural and tractable component of iterative AV development loops.

## 4.2 Results

Figure 3 presents retention-recall plots comparing BAMS, BAS, and baseline methods after each batch. BAMS achieves the highest retention at every retention level. Analyzing the results by batch reveals interesting trends. Initially, in Batch 1 methods within the same fidelity level exhibit similar performance: BAS, MC-GP, and DS-GP perform similarly, as do BAMS and MCM-GP. This is due to the use of the same random samples for initialization. However, the impact of the acquisition function becomes evident in the second batch, where BAMS significantly outperforms MCM-GP, and BAS outperforms other single-fidelity methods. This highlights the importance of the acquisition function in guiding the search towards potential failure modes.

By the third batch, BAMS achieves $80\%$ recall at a retention of $5p_\gamma N$, exceeding other methods by over $10\%$ and highlighting its superior ability to discover failure modes. BAS achieves the best recall among single-fidelity methods, demonstrating the effectiveness of our acquisition function. Additionally, MCM-GP performs comparably to BAS, showcasing the benefit of the multifidelity approach. These results also reinforce the advantages of leveraging multiple fidelity levels for efficient discovery, as both MCM-GP and BAMS outperform their single-fidelity counterparts. Notably, CE displays worse recall than all of the GP and multifidelity methods, but, as CE is still an adaptive method, it outperforms the non-adpative MC and DS baselines.

To illustrate the diversity of samples selected by each method, we use a determinantal point process (DPP) [49] to sample five diverse run segments from the collection of selected points at the end of the third batch. Figure 4 compares these run segments for BAMS and MCM-GP. The BAMS run segments include features such as busy intersections, pedestrians, and cyclists, whereas the MCM-GP run segments just show diversity in the road type (single lane vs. multi-lane). Table 1 demonstrates the evolution of focus for each method across batches, similar to the synthetic example (Figure 2c). Notably, BAMS concentrates on low TTC areas, facilitating the discovery of diverse failure modes. While CE shows low average $f(x)$ values in the third batch (cf. Table 1), the low recall performance in Figure 3 illustrates a common pitfall of CE: it collapses to sampling a subset of failure regions.

Now we turn to IS statistics: Figure 5 shows rate estimates and Table 2 shows recall and relative variance (RV). BAMS achieves the best recall while exhibiting a similar rate precision to BAS. These results reinforce the superior performance of multifidelity versions of each model. Additionally, GP-based methods demonstrate better recall and rate estimation compared to DS, CE, and MC.

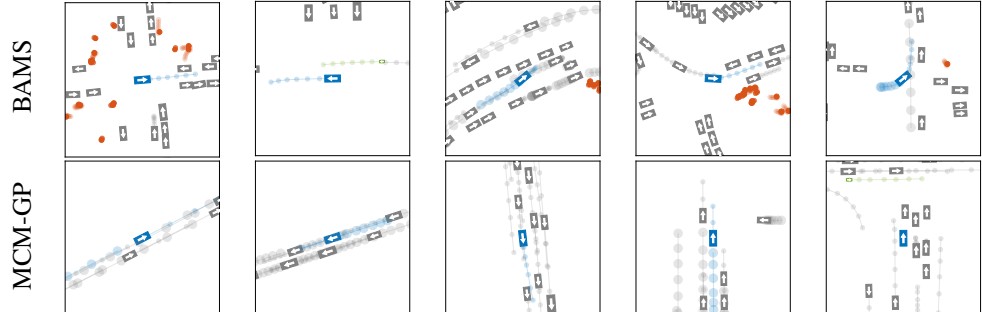

**Figure 4:** We select five samples from Batch 3 data using a DPP sampler to maximize diversity. The ego vehicle is blue and the snapshot corresponds to the timestamp at which the minimum TTC occurs; large transparent circles indicate the trajectory history, and small circles represent the future trajectory. The BAMS samples showcase busy intersections, pedestrians, and cyclists, whereas MCM-GP samples have less diversity.

| Method | Batch 1 | Batch 2 | Batch 3 |
|--------|---------|---------|---------|
| BAMS | **4.99 ± 0.51** | **1.75 ± 0.16** | **1.49 ± 0.21** |
| BAS | 6.55 ± 0.78 | 2.27 ± 0.58 | 3.19 ± 0.89 |
| MC-GP | 6.55 ± 0.78 | 6.34 ± 0.83 | 4.53 ± 0.6 |
| MCM-GP | **4.99 ± 0.51** | 6.09 ± 0.51 | 5.86 ± 0.55 |
| DS-GP | 6.55 ± 0.78 | 6.47 ± 0.81 | 7.37 ± 0.77 |
| CE | 6.55 ± 0.78 | 2.91 ± 0.26 | 2.20 ± 0.12 |

**Table 1:** The average $f(x)$ values and standard errors $(SE)$ for selected samples across all three batches. Analogous to Figure 2c, the transition from exploration to targeted selection is most evident for BAMS.

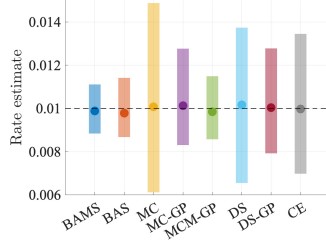

**Figure 5:** AV setting 90% confidence intervals for IS rate estimate with $K = 5p_\gamma N$ samples. The ground truth is $p_\gamma = 0.01$. Smaller error bars indicate a more precise estimator.

| Method | Recall $\pm SE$ | $100(RV \pm SE)$ |
|--------|-----------------|------------------|
| BAMS | **0.511 ± 0.0013** | **0.851 ± 0.121** |
| BAS | 0.388 ± 0.0013 | **0.969 ± 0.037** |
| MC | 0.048 ± 0.0006 | 13.5 ± 2.86 |
| MC-GP | 0.368 ± 0.0012 | 5.70 ± 2.39 |
| MCM-GP | 0.460 ± 0.0019 | 1.43 ± 0.288 |
| DS | 0.051 ± 0.0007 | 5.40 ± 0.096 |
| DS-GP | 0.315 ± 0.0012 | 3.67 ± 0.599 |
| CE | 0.096 ± 0.0007 | 4.43 ± 0.063 |

**Table 2:** IS evaluation for AV experiments. Recall and relative variance of rate estimates are calculated after Batch 3 with $K = 5p_\gamma N$ samples. Standard error (SE) is over 10 seeds.

## 5 Limitations

While BAMS demonstrates substantial improvements for efficiently discovering novel, high-likelihood scenarios and estimating the rate of adverse events, our approach has limitations. Specifically, to ensure tractable GP inference, we must perform dimensionality reduction on our input embeddings; this pre-training step adds computational overhead. Furthermore, the effectiveness of BAMS hinges on the availability of a continuous (or at least real-valued) metric function $f(x)$; this limits applicability in scenarios with purely binary labels. Finally, despite using an adaptive approach for selecting evaluation points across batches, our final IS step for rate estimation is not adaptive. Incorporating adaptive mechanisms within the IS stage, potentially leveraging techniques from sequential Monte Carlo (SMC) estimation, promises further enhancements to both the efficiency and accuracy of rate estimation.

## 6 Conclusion

This paper introduced *rate-informed discovery* to address the critical need for efficient estimation of AV safety and discovery of high-impact failure cases. We proposed BAMS, a method that utilizes Bayesian adaptive sampling and multifidelity simulation. Our results demonstrated the effectiveness of BAMS in discovering novel, high-likelihood scenarios of undesirable AV behavior while simultaneously estimating the rate of adverse events. In future work, we plan to "close the loop" by embedding rate-informed discovery via BAMS in an iterative AV development cycle and measure its effectiveness at reducing the end-to-end cost and latency of planner improvement.

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

# A Algorithmic details

## A.1 BAMS algorithm

For notational brevity in Algorithm 1, we use $\Delta J_j(y_l) := J\left(\{(x,l)_j\}_{n+1}^{i-1} \cup \{(x,l)\}\right) - J\left(\{(x,l)_j\}_{n+1}^{i-1}\right)$.

---

**Algorithm 1** BAMS

---

   **input:** data points $\{(y_l)_i\}_{i=1}^{N}$ for all $l \in \{0, 1, \dots L\}$, posterior $\theta_n$, number of clusters $S$, budget $m$, overbudget parameter $\eta$
   Create $S$ clusters of points (Algorithm 2)
   **for** $s = 1$ **to** $S$
      $\text{cost}_s \leftarrow 0$
      $\text{Queue}_s \leftarrow [\emptyset]$
      **while** $\text{cost}_s < \lceil \eta m N_s / N \rceil$
         $y_l \leftarrow$ arg min of optimization problem (5) in cluster $s$
         $\text{cost}_s \leftarrow \text{cost}_s + c(y_l)$
         Store $(y_l, \Delta J_j(y_l), c(y_l))$ in $\text{Queue}_s$
   $\text{cost}_g \leftarrow 0$
   $\text{Set}_g \leftarrow \{\emptyset\}$
   **while** $\text{cost}_g < m$
      $\text{Candidates} \leftarrow [\,\emptyset\,]$
      **for** $s = 1$ **to** $S$
         $(y_l, \Delta J_j(y_l), c(y_l)) \leftarrow$ next item in $\text{Queue}_s$
         **if** $\text{cost}_g + c(y_l) < m$
            $\text{Candidates} \leftarrow \text{Candidates} + (y_l, \Delta J_j(y_l), c(y_l))$
      **if** $\text{Candidates} = [\,\emptyset\,]$ **break**
      $(y_l, \Delta J_j(y_l), c(y_l))_{\text{next}} \leftarrow \arg\min_{\text{Candidates}} \Delta J_j(y_l)$
      $\text{cost}_g \leftarrow \text{cost}_g + (c(y_l))_{\text{next}}$
      $\text{Set}_g \leftarrow \text{Set}_g + \{(y_l)_{\text{next}}\}$
   **return** the set of next points to evaluate $\text{Set}_g$

---

## A.2 Computation of Equation (4)

As noted in Proposition 1 of Chevalier et al. [46], we have

$$\beta_n\left(x; \{x_j\}_{n+1}^{n+m}\right) := E_{\theta_n}\left[h_{n+m}(x)|\{X_j = x_j\}_{n+1}^{n+m}\right]$$
$$= \Phi_2\left(\begin{pmatrix} s(x) \\ -s(x) \end{pmatrix}, \begin{pmatrix} t(x) & 1 - t(x) \\ 1 - t(x) & t(x) \end{pmatrix}\right),$$

where

$$s(x) := \frac{\gamma - \mu_n(x)}{\sigma_{n+m}(x)},$$

$$t(x) := \frac{\sigma_n^2(x)}{\sigma_{n+m}^2(x)} = 1 + \frac{\text{cov}_n(x, X_m)\text{cov}_n(X_m, X_m)^{-1}\text{cov}_n(X_m, x)}{\sigma_{n+m}^2(x)},$$

and $X_m \in \mathbb{R}^{m \times d}$ is the matrix of $m$ new evaluation points. Departing from the form in Chevalier et al. [46], we further simplify this result to a form that eliminates $\sigma_{n+m}$. In particular, note that we can rewrite

$$\hat{t}(x) := \frac{1}{t(x)} = 1 - \frac{\text{cov}_n(x, X_m)\text{cov}_n(X_m, X_m)^{-1}\text{cov}_n(X_m, x)}{\sigma_n^2(x)}.$$

We can then rewrite

$$\Phi_2\left(\begin{pmatrix} s(x) \\ -s(x) \end{pmatrix}, \begin{pmatrix} t(x) & 1 - t(x) \\ 1 - t(x) & t(x) \end{pmatrix}\right) = \Phi_2\left(\begin{pmatrix} \hat{s}(x) \\ -\hat{s}(x) \end{pmatrix}, \begin{pmatrix} 1 & \hat{t}(x) - 1 \\ \hat{t}(x) - 1 & 1 \end{pmatrix}\right),$$

where $\hat{s}(x) := \frac{\gamma - \mu_n(x)}{\sigma_n(x)}$. By computing the Cholesky decomposition of $\text{cov}_n(X_m, X_m)$ once (an $O(m^3)$ operation), we can then compute $\hat{t}(x_i)$ for all $i$ in $O(m^2 N)$ time by solving the triangular linear system. Then computing the expectation $E_X\left[\beta_n\left(X; \{x_j\}_{n+1}^{n+m}\right)\right]$ is $O(N)$, resulting in an overall complexity of $O(m^2 N)$.

## A.3 Recursive next point selection for expected point variance

We first outline the procedure for naive sequential selection followed by how we improve upon it.

Naive sequential selection goes as follows: given $q < m$ previously selected points, we select consider all $N - n - q = O(N)$ candidates for the next point. We compute the acquisition function (4) for all of them and choose the argument of the minimum, resulting in $O(N)$ iterations each of an $O(q^2 N)$ computation for a total of $O(q^2 N^2)$ cost. Since we must do this for $1 \leq q \leq m - 1$, the resulting naive sequential selection strategy is $O(m^3 N^2)$.

We increase efficiency over the naive strategy by exploiting the fact that we are simply adding a single point to the selected candidates, so we do not need to blindly perform matrix inversion on the entire $\text{cov}_n(X_{q+1}, X_{q+1})^{-1} k(X_{q+1}, x)$. Instead, we employ identities for block matrix inversion.

We introduce notation in this section purely for brevity; we do not use this notation anywhere else in the paper. Define the shorthand $\Sigma_{n,q+1} := \text{cov}_n(X_q, X_q)$. Then we can write (with the right hand side as shorthand),

$$\Sigma_{n,q+1} = \left( \begin{array}{c|c} \Sigma_{n,q} & \text{cov}_n(X_q, x_{q+1}) \\ \hline \text{cov}_n(x_{q+1}, X_q) & \text{cov}_n(x_{q+1}, x_{q+1}) \end{array} \right) =: \left( \begin{array}{c|c} A & f \\ \hline f & g \end{array} \right)$$

Defining shorthand $\text{cov}_n(X_{q+1}, x) := (k_q^T | k)^T$, we have via the block matrix inversion identity

$$\text{cov}_n(x, X_{q+1}) \Sigma_{n,q+1}^{-1} \text{cov}_n(X_{q+1}, x) = k_q^T A^{-1} k_q + \frac{1}{h}(k_q^T A^{-1} f)^2 - \frac{2k}{h}(k_q^T A^{-1} f) + \frac{k^2}{h}$$

where $h := g - f^T A^{-1} f$. Note that the first term on the right hand side is precisely $\text{cov}_n(x, X_q) \Sigma_{n,q}^{-1} \text{cov}_n(X_q, x)$, so we can define a recursive relationship as long as we can also write a recursive relationship for $k_q^T A^{-1} f$. For an arbitrary vector $(e_q^T | e)^T$ we can utilize the block matrix inversion identity again to show that

$$(e_q^T | e) \Sigma_{n,q+1}^{-1} \text{cov}_n(X_{q+1}, x) = e_q^T A^{-1} k_q + \left( \frac{e_q^T A^{-1} f}{h} - \frac{e}{h} \right) (k_q^T A^{-1} f - k),$$

where the first term on the right-hand-side is precisely $e_q^T \Sigma_{n,q}^{-1} \text{cov}_n(X_q, x)$. Then we have a recursive formula for

$$\text{cov}_n(x, X_{q+1}) \Sigma_{n,q+1}^{-1} \text{cov}_n(X_{q+1}, x)$$

which allows us to efficiently compute $\hat{t}(x)$ iteratively. In particular, we can compute $\hat{t}(x_i)$ for all $i$ in $O(N^2)$ time, after which we must also compute the acquisition function (4) for all $N - n - q$ candidates and choose the argument of the minimum, an $O(N^2)$ computation. Thus, the overall time complexity is $O(mN^2)$.

### A.4 Clustering algorithm

First we divide $P$ into $S$ independent clusters. A natural choice for this task is spectral clustering [50, 51], but this requires both computing and storing the entire posterior covariance matrix, which is expensive. Instead, we use the trained Matérn lengthscales at $l = 0$ from the GP model to rescale the input points $x_i$ and run $K$-means clustering on the $N$ points; Euclidean distance with the rescaled dimensions is a cheap approximation for the Matérn kernel. Since each point $x$ corresponds to $L$ points $y_l$, each resultant cluster has $N_s$ points at each of the $L$ levels, with $\sum_{s=1}^{S} N_s = N$ (for full details, see Algorithm 2).

Then, we solve the minimization problem (5) over the independent clusters, with each cluster given a budget proportional to its size $\lceil \eta m N_s / N \rceil$. Here, $\eta \geq 1$ is an "overbudget" parameter such that we draw more points than necessary after pooling together outputs from all clusters (cf. Algorithm 1).

The average complexity of $K$-means via Lloyd's algorithm is $O(SN)$ [52]; it is significantly faster than computing the acquisition function, yielding an overall speedup as shown in Section A.4. When there are $S$ equal-size clusters, the complexity of our sequential selection strategy is $O(\lceil \eta m / S \rceil N^2 / S)$. We can trivially take advantage of parallel computation in this approach as well.

As a heuristic to prevent extremely small cluster sizes, we perform a slightly modified version of $K$-means clustering. In particular, we first perform K-Means with $\hat{S} > S$ clusters. followed by iterative removal of the smallest clusters by merging them with another cluster with the smallest Haussdorff distance. For shorthand, let the point $z$ be the original data point $x$ scaled by the Matérn lengthscales at $l = 0$ from the GP model. Let $\mathcal{C}_i$ for the set of points in cluster with index $i$. Then, for points $z_a$ in cluster $i$ and $z_b$ in cluster $j$, let the Euclidean distance be denoted as $d(a, b)$. The

---

**Algorithm 2** Clustering with Haussdorff merges

---

**input:** data points $\{(z_i)\}_{i=1}^N$, desired number of clusters $S$, initial number of clusters $\hat{S}$
Create $\hat{S}$ clusters of points via K-means
**for** $i = 1$ **to** $\hat{S} - S$
    $s \leftarrow \arg\min_k |\mathcal{C}_k|$
    $j \leftarrow \arg\min_k d_H(\mathcal{C}_s, \mathcal{C}_k)$
    $\mathcal{C}_j \leftarrow \mathcal{C}_j \cup \mathcal{C}_s$
    Remove $\mathcal{C}_s$
**return** $S$ remaining clusters $\{\mathcal{C}_s\}_{s=1}^S$

---

Haussdorff distance between clusters $i$ and $j$ is defined as:

$$d_H(\mathcal{C}_i, \mathcal{C}_j) := \max\left(\max_{a \in \mathcal{C}_i} \min_{b \in \mathcal{C}_j} d(a, b) \,, \; \max_{b \in \mathcal{C}_j} \min_{a \in \mathcal{C}_i} d(a, b)\right) \tag{6}$$

The clustering procedure is shown in Algorithm 2.

## B  Technical Results

### B.1  Computation time for variance

The variance satisfies:

$$\begin{aligned}
\mathrm{Var}(\hat{p}_\gamma(\theta_n)) &:= E_{\theta_n}\left[(\hat{p}_\gamma(\theta_n) - E_{\theta_n}[\hat{p}_\gamma(\theta_n)])^2\right] \\
&\overset{(a)}{=} E_{\theta_n}\left[(\hat{p}_\gamma(\theta_n) - E_X[p_n(X)])^2\right] \\
&= E_{\theta_n}\left[(E_X[g(X, \theta_n) - p_n(X)])^2\right] \\
&= E_{\theta_n}\left[E_{X,X'}\left[(g(X, \theta_n) - p_n(X))(g(X', \theta_n) - p_n(X'))\right]\right] \\
&\overset{(b)}{=} E_{X,X'}\left[E_{\theta_n}\left[(g(X, \theta_n) - p_n(X))(g(X', \theta_n) - p_n(X'))\right]\right] \\
&= E_{X,X'}\left[\mathrm{Cov}(g(X, \theta_n), g(X', \theta_n))\right] \\
&\overset{(c)}{=} \frac{1}{N^2}\sum_{i,j=1}^N \mathrm{Cov}(g(x_i, \theta_n), g(x_j, \theta_n)),
\end{aligned}$$

where we assume Fubini's Theorem applies so that we may interchange expectations in equalities $(a)$ and $(b)$, and equality $(c)$ is due to the fact that $P$ is the empirical distribution function over $N$ samples $x_i$. The bivariate covariance above can be written as

$$\begin{aligned}
\mathrm{Cov}(g(X, \theta_n), g(X', \theta_n)) &= E_{\theta_n}\left[g(X, \theta_n)g(X', \theta_n)\right] - p_n(X)p_n(X') \\
&= \mathbb{P}\left(\theta_n(X) \leq \gamma, \theta_n(X') \leq \gamma\right) - p_n(X)p_n(X') \\
&= \Phi_2\left(\frac{\gamma - \mu_n(X)}{\sigma_n(X)}, \frac{\gamma - \mu_n(X')}{\sigma_n(X')}, \frac{\mathrm{cov}_n(X, X')}{\sigma_n(X)\sigma_n(X')}\right) - p_n(X)p_n(X'),
\end{aligned}$$

where $\Phi_2(a, b, r)$ is the cumulative distribution function for vector $(a, b)$ under the bivariate normal distribution

$$\mathcal{N}\left(\begin{pmatrix} 0 \\ 0 \end{pmatrix}, \begin{pmatrix} 1 & r \\ r & 1 \end{pmatrix}\right).$$

We implement an efficient numerical procedure for this bivariate normal integral via the Gaussian quadrature method outlined by Drezner [53], Drezner and Wesolowsky [54]. Computing $\mathrm{Var}(\hat{p}_\gamma(\theta_n))$ requires $\binom{N}{2}$ of these computations for $i \neq j$. When $i = j$, we can simplify $\mathrm{Cov}(g(x_i, \theta_n), g(x_j, \theta_n)) = p_n(x_i)(1 - p_n(x_i))$, and $p_n(x) = \Phi\left(\frac{\gamma - \mu_n(x)}{\sigma_n(x)}\right)$, where $\Phi(\cdot)$ is the cumulative distribution function for a standard (one-dimensional) normal. Overall, the computation time is dominated by the $\binom{N}{2}$ evaluations of $\Phi_2$, so the complexity is $O(N^2)$.

## B.2 Proof of Proposition 3.1

*Proof.* Let the probability distribution function be $\rho(x)$ such that $\int_{\mathcal{X}} \rho(x)dx = 1$. Then the variance satisfies:

$$\begin{aligned}
\mathrm{Var}(\hat{p}_\gamma(\theta_n)) &:= E_{\theta_n}\left[(\hat{p}_\gamma(\theta_n) - E_{\theta_n}[\hat{p}_\gamma(\theta_n)])^2\right] \\
&\overset{(a)}{=} E_{\theta_n}\left[(\hat{p}_\gamma(\theta_n) - E_X[p_n(X)])^2\right] \\
&= E_{\theta_n}\left[(E_X[g(X,\theta_n) - p_n(X)])^2\right] \\
&= E_{\theta_n}\left[\left(\int_{\mathcal{X}}(g(x,\theta_n) - p_n(x))\rho(x)dx\right)^2\right] \\
&\le E_{\theta_n}\left[\int_{\mathcal{X}}(g(x,\theta_n) - p_n(x))^2\rho(x)dx \int_{\mathcal{X}}\rho(y)dy\right] \\
&= E_{\theta_n}\left[\int_{\mathcal{X}}(g(x,\theta_n) - p_n(x))^2\rho(x)dx\right] \\
&= E_{\theta_n}\left[E_X\left[(g(X,\theta_n) - p_n(X))^2\right]\right] \\
&\overset{(b)}{=} E_X\left[E_{\theta_n}\left[(g(X,\theta_n) - p_n(X))^2\right]\right] \\
&= E_X\left[p_n(X)(1 - p_n(X))\right] \\
&= E_X\left[h_n(X)\right],
\end{aligned}$$

where the inequality is the Cauchy-Schwarz inequality, and we assume Fubini's Theorem applies so that we may interchange expectations in equalities $(a)$ and $(b)$. $\qquad\square$

## B.3 Proof of Corollary 3.2

*Proof.* The proof is immediate following Proposition 3.1. Namely,

$$\begin{aligned}
E_{\theta_n}\left[\mathrm{Var}(\hat{p}_\gamma(\theta_{n+m}))|\{X_j = x_j\}_{n+1}^{n+m}\right] &\overset{(a)}{\le} E_{\theta_n}\left[E_X\left[h_{n+m}(X)\right]|\{X_j = x_j\}_{n+1}^{n+m}\right] \\
&\overset{(b)}{=} E_X\left[E_{\theta_n}\left[h_{n+m}(X)|\{X_j = x_j\}_{n+1}^{n+m}\right]\right] \\
&= J\left(\{x_j\}_{n+1}^{n+m}\right),
\end{aligned}$$

where inequality $(a)$ follows from Proposition 3.1, and again we assume Fubini's Theorem applies so we may interchange expectations in equality $(b)$. $\qquad\square$

## C Parameter-tuning studies

Here we provide experiments which explore how we chose (i) the number of clusters, (ii) IS parameter $\alpha$ and (iii) cost for the cheap fidelity for simulation in our main experiments.

### C.1 Number of clusters

We explore the impact of varying the number of clusters on both computational time and recall achieved at a retention budget of $5p_\gamma N$ after Batch 3. Figure 6 illustrates the relationship between the number of clusters and the recall achieved at this retention level, averaged over 6 random seeds.

Based on these findings, our clustering approach can significantly speed up the process while maintaining an acceptable recall. We select $S = 6$ clusters for BAMS, achieving a substantial speedup of $17\times$ compared to the non-clustering approach while maintaining competitive recall performance.

### C.2 IS parameter $\alpha$

In BAMS, the importance sampler is defined as $p_n(x)^\alpha$. The parameter $\alpha$ plays a crucial role in aligning the IS with the objective of rate-informed discovery. We select $\alpha$ to minimize the relative variance of the rate estimate while maintaining acceptable recall for importance-sampled items. Figure 7 showcases the impact of changing $\alpha$ on both the relative variance of the rate estimate and the recall achieved by IS with budget of $5p_\gamma N$ samples. Based on this analysis, we choose $\alpha = 2.5$ as it minimizes the relative variance and achieves a satisfactory recall.

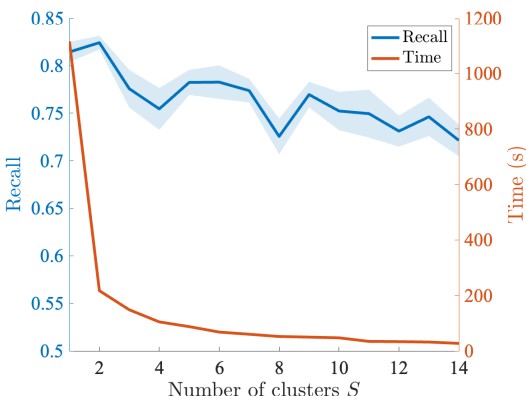

**Figure 6:** Exploring how the number of clusters affects both recall and computational time of BAMS's sequential selection stage at $5p_\gamma N$ retention. Recall is shown in blue, time in red.

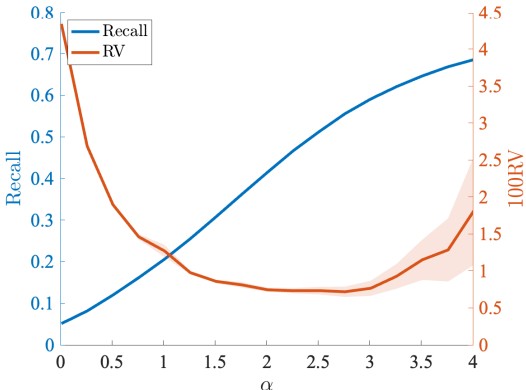

**Figure 7:** Influence of IS parameter $\alpha$ on relative variance of rate estimate and recall achieved using a budget of $5p_\gamma N$ samples. Recall is shown in blue, RV in red.

### C.3 Multifidelity cost

To determine the optimal level $l = 1$ cost in the AV setting, we conducted a parameter-tuning study. We varied the number of AV planner rollouts used to calculate the $Q_{0.25}(mTTC_{[1]} \ldots mTTC_{[i]})$ and evaluated BAMS recall at a retention budget of $5p_\gamma N$ after Batch 3. The cost factor for level 1 was set to $i/32$, where $i$ represents the number of rollouts used. We employed the same parameters as our main experiment, with $m_b = 15$ and $m_1 = 20$, and averaged the results across 6 random seeds.

Figure 8 depicts the recall achieved at retention level of $5p_\gamma N$ for varying level 1 costs. The results demonstrate that BAMS benefits from utilizing more samples, even with the added noise introduced by using fewer rollouts at level 1. The model can effectively learn the noise characteristics and achieve improved performance. Based on these findings, we selected $i = 5$ (and $c(y_1) = 5/32$).

## D Synthetic Experiments

**Objective** We consider a two-dimensional input $\mathcal{X} = \mathbb{R}^2$ and $P$ is the empirical distribution of $N = 20000$ samples from a Gaussian distribution $\mathcal{N}(0, I)$. Denoting $x_{[i]}$ as the $i^{\text{th}}$ dimension of $x \in \mathcal{X}$, our objective function is $f(x) = \left\| \left( \left| x_{[0]} \right| - c, x_{[1]} - c \right) \right\|_1$, where $c = 1.95$ determines the centers $(\pm c, c)$ of two diamonds for the failure modes. These modes are areas where $f(x) \leq \gamma$ with $\gamma = 0.56$, resulting in a rate $p_\gamma = 0.005$. Figure 2b visualizes $P$ and highlights failure modes.

**Multifidelity setup** Level 0 represents $f(x)$ without noise. Level 1 has a cost $c(y_1) = 0.10$ and adds Gaussian noise $\mathcal{N}(0, 0.1^2)$ to $f(x)$. For the initial batch (Batch 1), we employ a budget of $m_1 = 10$ evaluations. Subsequently, for batches 2 and 3, we allocate a budget of $m_b = 5$ evaluations across both fidelity levels.

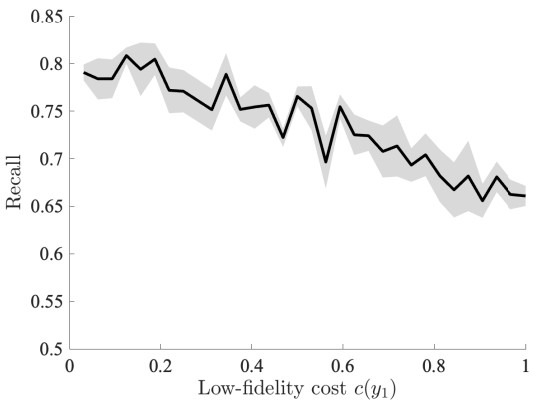

**Figure 8:** Impact of varying the cost of the lowest fidelity level (level 1) on recall at $5p_\gamma N$ retention after Batch 3. The data points represent the average recall across 6 random seeds.

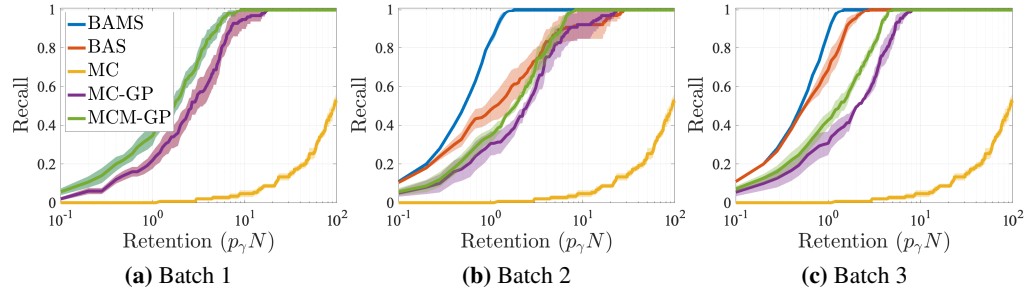

| (a) Batch 1 | (b) Batch 2 | (c) Batch 3 |

**Figure 9:** Retention-recall curves for synthetic experiments. The $x$-axis represents the retention budget scaled by $p_\gamma N$ while the $y$-axis shows the recall at each corresponding retention budget.

**Results** To illustrate the performance of our acquisition function, Figure 2c visualizes the samples selected by BAS. For clarity in the plot, we use a budget $m_b = 50$ (and we keep $m_1 = 10$). After the initial 10 random samples, the acquisition function guides exploration towards the top-left and top-right areas in Batch 2, revealing potential regions of interest. Finally, in Batch 3, it samples the boundaries of the set of failure modes, precisely targeting those points with the highest uncertainty. This visualization emphasizes the effectiveness of our acquisition function in progressively refining its focus, transitioning from initial exploration to targeted discovery.

Figure 9 showcases BAMS's superior performance in discovering failure modes. It consistently achieves the highest recall across all batches, surpassing other methods. Multifidelity approaches (BAMS and MCM-GP) outperform their single-fidelity counterparts (BAS and MC-GP), highlighting the advantage of leveraging different fidelity levels for efficient discovery. While BAMS and BAS achieve comparable performance in rate estimation (Figure 10), BAMS stands out with the highest recall (Table 3), signifying its exceptional ability to discover failure modes while accurately estimating the rate. This demonstrates BAMS's superior performance in rate-informed discovery. In Figure 10 we utilize $K = 2p_\gamma N$ samples for IS rather than $5p_\gamma N$ as in the AV experiments.

## E  Setup for the cross-entropy method

Our setup is as follows: we implemented the Cross-Entropy (CE) method using a multivariate Gaussian sampling distribution with a diagonal covariance matrix. At every iteration, we drew samples from the Gaussian and found the nearest neighboring run segments according to the Euclidean embedding distance. From the samples in this batch, we used the samples with the 5 most dangerous TTC values to update the mean and diagonal covariance of the sampling distribution. After 3 batches (like all of our other methods), we used the final multivariate distribution function in the third batch to generate importance scores for all run segments and performed rate estimation with a sample size of $5p_\gamma N$ like the other methods.

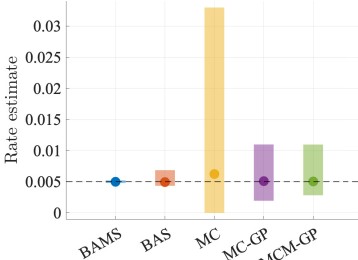

**Figure 10:** Synthetic setting $90\%$ confidence intervals for IS rate estimate with $K = 2p_\gamma N$ samples.

| Method | Recall $\pm SE$ | $100(RV \pm SE)$ |
|---|---|---|
| BAMS | **$1.00 \pm 0.000$** | **$2.00 \pm 0.003$** |
| BAS | $0.917 \pm 0.003$ | $3.85 \pm 0.366$ |
| MC | $0.010 \pm 0.001$ | $710. \pm 214.$ |
| MC-GP | $0.394 \pm 0.004$ | $39.4 \pm 3.88$ |
| MCM-GP | $0.537 \pm 0.005$ | $29.76 \pm 5.08$ |

**Table 3:** IS evaluation for synthetic experiments. Recall and relative variance of rate estimates are calculated after Batch 3 using IS with $K = 2p_\gamma N$ samples. Standard error (SE) is calculated over 10 seeds.

## F    Connections to other rare-event simulation and AV safety-evaluation techniques

Our work on rare event discovery and rate estimation complements several areas within the field of AV safety evaluation. For instance, while we focus on selecting informative run segments from a given set, other works like Feng et al. [55] address the orthogonal problem of efficient IS within a single run segment, often employing techniques like adversarial agents. Similarly, while methods like Arief et al. [56] rely on specific assumptions about the failure region's topology (e.g., orthogonal monotonicity), our approach remains agnostic to such constraints, enhancing its applicability to a wider range of scenarios.

Our work aligns with the broader themes of scenario generation and safety evaluation. Existing surveys offer comprehensive overviews of scenario generation methods [57, 58, 59], categorizing our approach under data-driven or learning-based techniques. Importantly, our method addresses the crucial challenge of transferability, highlighted in Ding et al. [57], by focusing on efficiently identifying critical scenarios within a given logged dataset rather than relying on potentially brittle synthetic scenario generation.

This work was motivated by the desire to develop a Bayesian formulation for Sequential Monte Carlo (SMC). While non-Bayesian SMC approaches exist [35, 37], a Bayesian perspective offers several benefits. Our primary contribution, BAMS, embodies the novel Bayesian component of this broader research direction and is designed for seamless integration into SMC frameworks.

A key advantage of a Bayesian formulation lies in reducing the sample complexity of Markov Chain Monte Carlo (MCMC) steps within the sequential levels of SMC. Specifically, by leveraging a GP, BAMS performs these MCMC steps directly within the embedding space, completely bypassing the need for simulations during this phase. This approach stands in contrast to previous suggestions in [35] like using surrogate simulations for a portion of the MCMC steps followed by a final simulation-based Metropolis-Hastings (MH) acceptance step. In particular, the Bayesian formulation requires no simulation for any of the MH steps, contributing significantly to its efficiency.

**Theoretical bounds on multilevel splitting performance**    We can quantify the intuition of MCMC being expensive by comparing the theoretical lower bound of simulations needed by (single fidelity) multilevel splitting to achieve the same relative variance as BAMS and BAMS. Using the notation in Norden et al. [37], the number of iterations is $K = \log(p_\gamma)/\log(1-\delta)$, the total simulation cost is $N + T\delta NK$, and the resulting relative variance is $K\frac{\delta}{N(1-\delta)}$. Using the standard value of $\delta = 0.1$ for splitting and the desired relative variance of $0.00851$ (the value for BAMS after Batch 3), we can calculate $K = 43$, $N = 561$, and the total number of simulations is $561 + 2412T \geq 2973$, where the lower bound ($T = 1$) is achieved only if we use surrogate simulation to eliminate all but one of the iterations of simulation during MCMC. BAMS only has 2296 simulations, so vanilla multilevel splitting requires at least $30\%$ more simulation cost for the same precision. Carrying out the same math for BAS (with relative variance $0.00969$), we get $N = 493$ and multilevel splitting has to have at least 2613 simulations for the same precision. Fixing the budget for multilevel splitting to 2296 simulations, multilevel splitting has a lower bound of $100RV \geq 1.10$, which is worse than BAMS and BAS.

How loose are these lower bounds for simulation cost given a fixed precision, and a fixed simulation cost? The discrete structure of our problem space (since we consider the base distribution $P$ as a discrete distribution over $N$ run segments) creates difficulty for implementing vanilla SMC baseline approaches. In particular, efficient MCMC techniques such as Hamiltonian Monte Carlo cannot be easily utilized. In this sense, the theoretical lower bounds are quite loose because it is likely that we need to perform more than one MH step using the real simulator. Furthermore, creating competitive SMC baseline approaches would require developing a novel graph-based MCMC kernel (to efficiently deal with the discrete structure); this is out of scope for this paper and may be a novel paper itself. Part of the motivation for this paper is to develop a Bayesian approach that sidesteps the necessity of performing difficult MCMC on a discrete space.

**Comparison with the cross-entropy method** In addition to the advantages of BAMS over SMC methods, it also demonstrates superior performance compared to the cross-entropy method [47], as detailed in Section 4.2. BAMS bears many similarities to the cross-entropy method. In particular, we have a sampling distribution (built upon the GP in our case) that updates as we iteratively sample. In BAMS, we utilize an optimization problem to perform sampling rather than simply drawing from a parametric sampler.

