# OpenReview forum: "Rate-Informed Discovery via Bayesian Adaptive Multifidelity Sampling"
_robot-learning.org/CoRL/2024/Conference — CoRL 2024_

### Official Review · Reviewer_GzcW · 2024-07-10
**Review of Submission379**

**Originality:** 2
**Technical Quality:** 3
**Clarity Of Presentation:** 3
**Potential Impact:** 2
**Recommendation:** 3
**Confidence:** 4

**Review:**

**Strength**

1. The motivation for estimating AV performance under risky scenarios and discovering potential failure cases is promising. This is definitely an unsolved but crucial problem in the autonomous driving field.
2. This paper is mostly well-written and well-organized. It is easy to understand the core idea and main contribution.

**Weakness**

1. A lot of details are missing. Some examples can be found in my questions. Overall, I think some of the concepts mentioned in the paper may be familiar to audients with a statistical background rather than a robotics background. Some terms also have different meanings in these two domains, for example, embedding model and simulator.
2. Novelty is a little bit limited as estimating the rate of rare events in autonomous driving has been investigated in a lot of previous work. MC and IS are two well-known methods but I think there are some more recent works in this direction.
3. Based on the previous point, there is a lot of missing literature on rare-event simulation for autonomous driving, for example, [1-6]. I think the authors should thoroughly discuss the connection between the proposed method and these existing methods. Comparison in the experiment section is also necessary.
4. In addition to the estimation of the rate of rare events, I think this paper also proposes to discover failure scenarios. Then, safety-critical driving scenario generation is also a large area that needs to be included in the related work section. For example, the authors can start with some survey papers [7-9].

---
[1] Feng, S., Sun, H., Yan, X., Zhu, H., Zou, Z., Shen, S., & Liu, H. X. (2023). Dense reinforcement learning for safety validation of autonomous vehicles. Nature, 615(7953), 620-627.

[2] Sinha, Aman, et al. "Neural bridge sampling for evaluating safety-critical autonomous systems." Advances in Neural Information Processing Systems 33 (2020): 6402-6416.

[3] Norden, Justin, Matthew O'Kelly, and Aman Sinha. "Efficient black-box assessment of autonomous vehicle safety." arXiv preprint arXiv:1912.03618 (2019).

[4] O'Kelly, M., Sinha, A., Namkoong, H., Tedrake, R., & Duchi, J. C. Scalable end-to-end autonomous vehicle testing via rare-event simulation. NeurIPS 2018.

[5] Arief, M., Huang, Z., Kumar, G. K. S., Bai, Y., He, S., Ding, W., ... & Zhao, D. (2021, March). Deep probabilistic accelerated evaluation: A robust certifiable rare-event simulation methodology for black-box safety-critical systems. In International Conference on Artificial Intelligence and Statistics (pp. 595-603). PMLR.

[6] Arief, M., Huang, Z., Kumar, G. K. S., Bai, Y., He, S., Ding, W., ... & Zhao, D. (2020). Deep probabilistic accelerated evaluation: A certifiable rare-event simulation methodology for black-box autonomy. arXiv preprint arXiv:2006.15722, 1(4).

[7] Ding, W., Xu, C., Arief, M., Lin, H., Li, B., & Zhao, D. (2023). A survey on safety-critical driving scenario generation—A methodological perspective. IEEE Transactions on Intelligent Transportation Systems, 24(7), 6971-6988.

[8] Zhong, Z., Tang, Y., Zhou, Y., Neves, V. D. O., Liu, Y., & Ray, B. (2021). A survey on scenario-based testing for automated driving systems in high-fidelity simulation. arXiv preprint arXiv:2112.00964.

[9] Schütt, B., Ransiek, J., Braun, T., & Sax, E. (2023, June). 1001 ways of scenario generation for testing of self-driving cars: A survey. In 2023 IEEE Intelligent Vehicles Symposium (IV) (pp. 1-8). IEEE.

**Quality Of The Limitations Section:**

2

**Questions For Rebuttal:**

1.	Figure 1 shows the pipeline of a loop of finding adversarial scenarios and using it to improve the planner. However, I couldn’t find the stage of the planner improvement in this paper.
2.	What does “reducing the end-to-end cost” mean in the last sentence of the conclusion section?
3.	In line 227, what is the planner used in the sentence “32 rollouts per run segment generated via AV planner simulation”? Why are these 32 rollouts different?
4.	In line 230, why use an embedding model to generate x? What is the specific definition of x rather than “the realization of the AV and its environment”?
5.	I am not sure about how the multi-fidelity setting is implemented during the experiment. Does it mean we use different planners to simulate the same initial positions, or do we use different simulators (e.g., Carla, highway-env, Waymax, …)? I think the terminology may lead to some ambiguity here.
6.	What is the connection between “simulators” and “ﬁdelities”?
7.	In line 246, why use $\gamma=0.43$ as the threshold?
8.	In Figure 4, the authors show some examples of the discovered scenarios. The authors say that the scenarios discovered by BAMS showcase busy intersections, pedestrians, and cyclists. However, I think it is not enough to show that these scenarios are adversarial and interesting. It is also easy to use a rule-based filter to find those scenarios.

**Robotics Focus:**

2

**Summary Of Paper:**

This paper aims to accurately estimate the performance of AV under risk scenarios and efﬁciently discover potential failure cases. To do so, the authors introduce Bayesian adaptive multiﬁdelity sampling (BAMS), which prioritizes exploration of regions with potentially low performance, leading to the identiﬁcation of novel and critical scenarios that traditional methods might miss. Previous work such as naive Monte Carlo (MC) is intractably expensive for rate-informed discovery. Furthermore, importance sampling (IS) that rely heavily on previously determined failure modes are poor at discovering novel issues. Compared with these two methods, BAMS discovers 10 times as many issues as MC and IS methods. In addition, BAMS also generates rate estimates with variances 15 and 6 times lower than MC and IS respectively.

**Summary Of Recommendation:**

My major concern is that the novelty of this paper is limited due to omit of a lot of related works. In addition, the paper needs to be improved to include more details about some design choices and terminologies.

---

### Official Review · Reviewer_EQTe · 2024-07-30

**Originality:** 3
**Technical Quality:** 4
**Clarity Of Presentation:** 4
**Potential Impact:** 3
**Recommendation:** 3
**Confidence:** 3

**Review:**

1. Strengths/Weaknesses:

Strengths:

Innovative approach: The multifidelity approach of BAMS is novel and provides a balance between computational cost and accuracy by combining different levels of simulation fidelity.

Effective acquisition function: The proposed acquisition function effectively guides the exploration and discovery of failure modes, improving recall and reducing variance in rate estimates.

Strong empirical results: The experimental results show that BAMS outperforms existing methods in terms of recall and rate estimation accuracy in the multifidelity setting.

Weaknesses:

Additional implementation details and open-source code availability would improve reproducibility.

Limitations Section is not provided.

2. Clarity, Quality, Novelty And Reproducibility:

Clarity: The paper is well-written and clearly organized.

Quality: High-quality in terms of theoretical development and empirical validation.

Novelty: The multifidelity approach, the specific acquisition function, and the practical (clustering) relaxation used in BAMS are novel contributions to the field.

Reproducibility: While the theoretical aspects are well documented in the appendix, additional details on implementation and code availability would improve reproducibility.

3. Correctness:

The paper appears to be correct in its theoretical foundations and empirical validations.

The proofs and propositions presented support the claims made about the algorithm's performance.

4. Novelty And Significance:

Technical: BAMS represents an advancement in the field of autonomous vehicle simulation by introducing a method that efficiently combines multiple fidelities of data.

Empirical: The empirical results are compelling, showing that BAMS consistently outperforms existing methods in detecting diverse failure modes and providing accurate rate estimates.

This empirical strength underscores the practical utility and potential impact of the proposed method in AV development.

**Quality Of The Limitations Section:**

1

**Questions For Rebuttal:**

None

**Robotics Focus:**

2

**Summary Of Paper:**

This paper introduces the Bayesian Adaptive Multifidelity Sampling (BAMS) algorithm for improving rate-informed discovery in the context of autonomous vehicle (AV) simulations. BAMS uses multifidelity simulations to efficiently identify diverse failure modes by combining high-fidelity and low-fidelity data to improve the discovery and estimation processes.

**Summary Of Recommendation:**

Accept: The paper should be accepted for its novel approach, strong theoretical foundations, and impressive empirical results.

---

### Author Rebuttal · Authors · 2024-08-07

Please see our attached updated paper which addresses the concerns brought up by both reviewers. We have updated the background and literature review, updated the experiments with a further baseline method, the cross-entropy method, and added a limitations section. The Appendix contains further comparisons with Sequential Monte Carlo (SMC) methods, which we also detailed in our comment to Reviewer GzcW.

Thank you for your feedback and we welcome further discussion.

---

### Decision · Program_Chairs · 2024-09-04

**Decision:**

Accept

**Comment:**

The paper was initially evaluated with mixed reviews, but after the rebuttal, we the reviewers suggest accepting the paper.

Here is a high level overview of the reviews. (before rebuttal)

### Strengths:
1. The multifidelity approach of BAMS is novel and provides a balance between computational cost and accuracy by combining different levels of simulation fidelity. (Reviewer EQTe)
2. The proposed acquisition function effectively guides the exploration and discovery of failure modes, improving recall and reducing variance in rate estimates. (Reviewer EQTe)
3. The experimental results show that BAMS outperforms existing methods in terms of recall and rate estimation accuracy in the multifidelity setting. (Reviewer EQTe)
4. The motivation for estimating AV performance under risky scenarios and discovering potential failure cases is promising and crucial for the autonomous driving field. (Reviewer GzcW)

### Weaknesses:
1. **Lack of Implementation Details**: Additional implementation details and open-source code availability would improve reproducibility. (Reviewer EQTe)
2. The paper does not provide a limitations section, which is essential for a comprehensive understanding of the work. (Reviewer EQTe)
3. **Insufficient Related Work Discussion**: The paper lacks a thorough discussion of related works in the literature, especially on rare-event simulation for autonomous driving and safety-critical driving scenario generation. (Reviewer GzcW)
4. There are many ambiguous terms and missing details in the paper, such as the definitions of certain concepts and the specifics of the multifidelity setting implementation. (Reviewer GzcW)

The reviewers find their concerns to be sufficiently addressed. The authors also improved organization an readability in the latest version of the paper. I suggest accepting the paper.